# Evaluation of Bone Gain and Complication Rates after Guided Bone Regeneration with Titanium Foils: A Systematic Review

**DOI:** 10.3390/ma13235346

**Published:** 2020-11-25

**Authors:** Elisabet Roca-Millan, Enric Jané-Salas, Albert Estrugo-Devesa, José López-López

**Affiliations:** 1Faculty of Medicine and Health Sciences (School of Dentistry), University of Barcelona, 08907 Barcelona, Spain; erocamil@gmail.com; 2Oral Health and Masticatory System Group-IDIBELL, Faculty of Medicine and Health Sciences (School of Dentistry), Odontological Hospital University of Barcelona, University of Barcelona, 08907 Barcelona, Spain; enjasa19734@gmail.com (E.J.-S.); albertestrugodevesa@gmail.com (A.E.-D.)

**Keywords:** titanium membrane, titanium foil, occlusive titanium barrier, bone augmentation, guided bone regeneration

## Abstract

Guided bone regeneration techniques are increasingly used to enable the subsequent placement of dental implants. This systematic review aims to analyze the success rate of these techniques in terms of bone gain and complications rate using titanium membranes as a barrier element. Electronic and hand searches were conducted in PubMed/Medline, Scielo, Scopus and Cochrane Library databases for case reports, case series, cohort studies and clinical trials in humans published up to and including 19 September 2020. Thirteen articles were included in the qualitative analysis. Bone gain both horizontally and vertically was comparable to that obtained with other types of membranes more commonly used. The postoperative complication rate was higher that of native collagen membranes and non-resorbable titanium-reinforced membranes, and similar that of crosslinked collagen membranes and titanium meshes. The survival rate of the implants was similar to that of implants placed in native bone. Due to the limited scientific literature published on this issue, more randomized clinical trials comparing occlusive titanium barriers and other types of membranes are necessary to reach more valid conclusions.

## 1. Introduction

The four classical principles of guided bone regeneration (GBR) are primary closure, angiogenesis, space maintenance and blood clot stability [1] (Figure 1).

Based on these concepts, different techniques and a wide variety of biomaterials have been developed with the aim of achieving greater predictability, lower risk of complications, lower morbidity and shorter operative time in this type of treatment, which is becoming more and more common [2,3]. Research on new biomaterials for bone regeneration is advancing rapidly; even recently they have been manufactured by combining biopolymers and natural nanoparticles [4,5]. Ideally, the proposed method should provide a solution to the four precepts [2,3].

On the other hand, the properties that an ideal biomaterial should fulfill are osteogenesis, osteoconduction and osteoinduction. Therefore, autologous bone is considered the gold standard [2]. However, the great resorption, the unlimited availability, the morbidity and the longer surgical time represent inconveniences in its use. That is why a combination of several biomaterials is generally used in GBR procedures [2,3,6].

Since clot formation is the first and essential step in bone healing [7,8], in the last years, there have been numerous studies that focus on the use of blood concentrates (platelet-rich plasma (PRP), platelet-rich fibrin (PRF) and platelet-rich growth factor (PRGF)) in these surgical procedures [9,10,11].

However, due to the presence of erythrocytes, blood has a greater capacity to generate thrombin and activate platelets than these concentrates [12]. Likewise, the lower porosity and density of the fibrin layer present in the complete clot facilitate cell migration [13]. However, to benefit from its properties, it would be necessary to at least maintain the space and stabilize the clot [14,15,16].

To comply with these two principles, there are different types of membranes or barrier elements, such as titanium-reinforced polytetrafluoroethylene membranes, perforated titanium meshes and titanium foils [17]. The main drawback of the first two is the high exposure rate, associated with a high failure rate [3,6,18]. However, titanium barriers tolerate prolonged exposure to the oral environment, with good hygiene and the use of antiseptics to avoid bacterial colonization being essential [19,20,21].

Several studies published in recent years defend the use of these barrier elements in the regeneration of large maxillary atrophies [20,21,22,23,24], in post-extraction socket reconstruction [19,25,26], in the regeneration of periodontal defects [27] and even simultaneously with implant placement [28].

The concept of using these barriers is to take advantage of the properties of the blood clot, which is why, in some studies, they are used without biomaterial filling [21,24]; although, in other publications, these membranes have been used in combination with PRF [19], allograft [20,22,23,29], xenograft [25,26,30], mixed autograft and allograft [31] or even tricalcium β-phosphate [28].

Given that titanium barriers comply with the two aforementioned precepts and have tolerance to prolonged exposure, the objective of this systematic review is to study the success rate of GBR through the use of this type of membranes in terms of the amount of new bone formed and the complications associated with this surgical technique.

## 2. Materials and Methods

This systematic review was conducted according to the guidelines of the Preferred Reporting Items of Systematic Reviews and Meta-Analyses (PRISMA) statement [32]. Before starting the review, a detailed protocol of the methodology was developed. The protocol was not registered.

### 2.1. Focused Questions

Is the use of occlusive titanium barriers alone or in combination with biomaterial a predictable treatment in terms of amount of new bone formed? (primary question)What is the complication rate regarding membrane exposure and infection? (primary question)What is the survival and success rate of implants placed after this regenerative procedure? (secondary question)

### 2.2. PICO Question

P: Patients with partial or total edentulism.

I: Guided bone regeneration using occlusive titanium barriers alone or in combination with biomaterials.

C: Guided bone regeneration using other type of membranes.

O: Amount of new bone formed and rate of membrane exposure and infection.

### 2.3. Eligibility Criteria

Inclusion criteria: Case reports, case series, cohort studies and clinical trials written in English or Spanish that analyze the use of titanium foils in GBR procedures were considered for inclusion.

Exclusion criteria: animal studies. No limitations were used for publication date, sample size, follow-up period, type of bone defect treated or filler biomaterial.

### 2.4. Search Strategy

An electronic search was performed by two reviewers (E.R-M and E.J-S) for articles published up to and including September 2020. The databases consulted were MEDLINE/PubMed, Scielo, Scopus and Cochrane Library. An additional hand search was conducted to identify potential articles of interest in the references of the studies found. Both searches were performed on 19 September 2020.

The following term combination was used in the electronic search: (“titanium membrane [All Fields]” OR “occlusive titanium barrier [All Fields]” OR “titanium foil [All Fields]”) AND (“bone regeneration [All Fields]” OR “bone formation [All Fields]” OR “bone augmentation [All Fields]” OR “guided bone regeneration [All Fields]” OR “guided tissue regeneration [All Fields]”).

### 2.5. Study Selection

After screening titles and discarding duplicates, those studies whose abstract met the inclusion criteria were selected. The full text of these articles was read to verify that they met the eligibility criteria. Disagreements during the study selection were solved by consulting a third author (J.L-L).

### 2.6. Data Extraction and Method of Analysis

The data were extracted by two authors (E.R-M and A.E-D) and entered into a data collection form (Microsoft Excel version 16.35). In case of disagreement, a third author (E.J-S or J.L-L) was consulted, to get a consensus. The following data were collected: author(s), year of publication, type of study, number of titanium foils, type of defect, filling material, time of membrane removal, amount of bone gain, percentage membrane exposure, percentage of infection, number of patients, number of implants placed, survival and success rates of the implants, and follow-up period. Corresponding authors were contacted and asked if they could provide missing data. To summarize the data, the mean rates of exposure, infection, implant survival and implant infection were calculated.

### 2.7. Quality Assessment and Risk of Bias

The Strength of Recommendation Taxonomy (SORT) criteria were used to assess the quality of the evidence provided in the included studies [33]. This classification groups the articles into three levels: Level 1 (good-quality patient-oriented evidence), Level 2 (limited-quality patient-oriented evidence) and Level 3 (other evidence). Version 2 of the Cochrane Collaboration’s tool for assessing risk of bias in randomized trials (RoB 2) [34] was implemented to evaluate the risk of bias of the randomized clinical trials and the Cochrane tool for assessing risk of bias in non-randomized studies of interventions (ROBINS-1) [35] was implemented to assess the risk of bias of the included retrospective cohort study.

## 3. Results

### 3.1. Study Selection

Through the electronic and manual searches, a total of 118 records were identified. After reading the titles and, if necessary, the abstracts and identifying duplicate articles, a total of 101 papers were discarded. Of these 17 studies assessed for eligibility, one was discarded after reading the full text because the titanium foil was placed in order to stabilize the jaw and prevent its fracture [36]. Three other articles were discarded, as they appear to be the same study as another two of the included papers but in earlier stages [20,21,23]. A total of 13 articles were included in the qualitative analysis [19,22,24,25,26,27,28,29,30,31,37,38,39] (Figure 2). A quantitative analysis could not be performed due to the lack of information provided and the great heterogeneity of the studies, in terms of sample size, type of defect to be regenerated, time of membrane removal and filling material used.

### 3.2. Study Methods and Characteristics

Two of the included articles were case reports [28,30], seven were case series [19,22,25,26,29,31,39], three were split-mouth randomized clinical trials [27,37,38] and only one was a retrospective cohort study [24]. The studies were published between 1999 and 2020 (Figure 3 and Table 1).

The total population was 200 patients (72 women, 54 men, and 74 not specified), in which 260 titanium foils were placed. In two of the studies, the membrane was custom-fit manufactured from a previous computerized tomography [22,24]. The follow-up period was between two months and nine years. In six of the included studies [19,22,24,26,28,30], implants were placed in the regenerated area, with 88 implants placed in 39 patients. The time of membrane removal was between 21 days and 17 months; although, in some studies, it is not specified [19,28,31]. 

A wide variety of bone defects were treated: horizontal [25,28,29,30,39], horizontal and vertical [22,24,26,28,39], periodontal intrabony defects [27,37], post-extraction sockets [19,38], peri-implant defects [31,39] and sinus-floor augmentation [39].

The filling material was different depending on the study: blood clot [24,27,37,38], xenograft [25,26,30], allograft [22,29], β-Tricalcium phosphate [28], PRF or PRF + tricalcium phosphate + hydroxyapatite [19], autograft [38], autograft or/and hydroxyapatite [39] and autograft + xenograft [31].

### 3.3. Quality Assessment and Risk of Bias

According to the Strength of Recommendation Taxonomy (SORT) criteria, four of the studies included obtained a Level 2 of evidence [24,27,37,38] while the others obtained a Level 3 [19,22,25,26,28,29,30,31,39], so it should be noted that the quality of the included studies is limited and greater value should be given to the results obtained in Level 2 articles. Table 2 presents the risk of bias of the randomized clinical trials (RCT) [27,37,38] and the cohort study [24] assessed by using RoB 2 and ROBINS-1, respectively, with an overall judgment of low–moderate risk of bias. The present review itself fulfils 22 items in the PRISMA statement [32]. 

### 3.4. Bone Gain

Of the 13 studies included, four did not specify or evaluate the amount of new bone formed [19,24,27,28]. In two of them, a sufficient amount of bone was gained in all cases [19,28]. Another, which studied the regeneration of periodontal intrabony defects, did not evaluate this parameter [27]. The other one did not specify the values obtained but clarified that, in most cases, a sufficient bone augmentation was obtained for implant placement [24].

In the other nine studies, in which a wide variety of bone defects were treated, considerable bone gain was obtained after using titanium foils, regardless of the type of defect. Vertical bone gain, excluding the post-extraction sockets, was between 4.5 mm (autograft + xenograft) [31] and 7.3 mm (xenograft) [26]. Horizontal bone gain was between 2.3 mm (allograft) [29] and 9 mm (xenograft) [25]. In the study that quantified the regeneration of periodontal intrabony defects, a filling of 54.69% was obtained [37].

### 3.5. Complications

In two of the studies, titanium barriers were intentionally left exposed [25,26]. Three other articles reported not having had any exposure prior to membrane removal [26,27,29]. In the rest of the studies, the exposures ranged between 21.43% and 50% [19,22,24,27,31,37,38,39]. The mean percentage of accidental exposure was 23.81%. Only one article reported having cases of graft infection, with a percentage of 11.9% [31].

It must be taken in account that one of these articles reported an exposure of the 43.75% of the membranes; however, this percentage is higher, since the study discarded those patients who had had an exposure greater than a quarter of the membrane in the first four to six weeks [27].

As another complication, in one of the studies two patients were excluded due to displacement of the titanium barrier [37].

### 3.6. Implant Survival and Success Rates

In six of the included studies [19,22,24,26,28,30], implants were placed in the regenerated area, with 88 implants placed in 39 patients. The mean survival rate was 96.5% (82.6–100%), and the mean success rate was 91.3% (82.6–100%). It must be considered that only two studies evaluated the success rate [22,26]. The follow-up period for these implants was between one and nine years (Table 3). The study with the lowest survival rate was the one with the longest follow-up period [24].

## 4. Discussion

According to the results obtained in the present systematic review, the horizontal bone gain was between 2.3 and 9 mm [22,25,26,29,30], and the vertical between 4.5 and 7.3 mm [22,26,31]. These last values without taking into account the randomized clinical trial in which alveolar ridge preservation of well-conserved post-extraction sockets was performed, in which the mean vertical gain was greater than 8 mm [38]. 

In a recent RCT comparing vertical bone gain by using d-PTFE titanium-reinforced membranes or titanium meshes, a gain of 4.2 ± 1.0 mm (range 2.7–5.8) and 4.1 ± 1.0 mm (range 2.6–6.3) was obtained, respectively [40]. Likewise, a meta-analysis obtained similar results, with a mean vertical bone gain of 4.42 mm by using non-resorbable membranes (d-PTFE and e-PTFE), of 4.26 mm by using titanium meshes covered by resorbable membranes and of 5.2 mm by using titanium meshes alone [41].

Based on these data, it appears that the use of titanium foils is predictable in terms of the amount of bone gain, regardless of the filling material, and the gain may be even higher than with the use of other commonly used non-resorbable membranes or meshes.

With regard to horizontal bone gain, the values obtained in the different studies analyzed are very heterogeneous and do not seem to be related to the filling material used either. Other studies in which horizontal regeneration procedures were performed with collagen membranes and particulate grafts obtained average bone gains of 2.27 ± 1.68 mm [42], 5.68 ± 1.42 mm [43] and 5.03 ± 2.15 mm [44]. Thus, it seems that, in terms of horizontal bone gain, titanium barriers are comparable to collagen membranes, the most widely used in horizontal ridge augmentation procedures.

Based on the included articles, there is no evidence to believe that a filling material is better than another or even blood clot, in combination with occlusive titanium barriers. Furthermore, this type of membrane could be useful in the regeneration of defects of different types, from contained defects such as a post-extraction socket to a combined vertical and horizontal defect such as a posterior mandibular atrophy.

Regarding complications, it appears that titanium foils are prone to exposure, as is the case of titanium meshes and non-resorbable membranes with titanium reinforcement. The mean exposure rate in the present work was 23.81% (range 0–50%) [19,22,24,27,28,29,30,31,37,38,39], and the mean infection rate was 1.19% (range 0–11.9%) [19,22,25,26,28,29,30,31,37,38]. If these results are compared with those of other studies, it can be observed that the rate of postoperative complications of titanium foils is slightly higher than that of GBR procedures with other types of membranes. In an RCT in which the rate of complications in vertical ridge augmentation was evaluated through the use of titanium meshes covered with collagen membranes and the use of non-resorbable membranes with titanium reinforcement, a postoperative complication rate (exposure and infection) of 21.1% and 15% was obtained, respectively [40]. A meta-analysis obtained an intra- and postoperative complications rate of 21% for titanium meshes covered with resorbable membranes, of 6.9% for non-resorbable membranes and of 20% for titanium meshes [41]. 

In different studies on the use of native collagen membranes in horizontal bone regeneration, a percentage of complications of 3.2% [43] and 0% [44] was recorded. In a recent meta-analysis, an exposure rate of 28.62% for crosslinked membranes and of 20.74 for non-crosslinked membranes was obtained [42].

The postoperative complication rate of titanium barriers was higher than that of native collagen membranes and non-resorbable titanium-reinforced membranes, and similar to that of crosslinked collagen membranes and titanium meshes.

It must be taken into account that the complication rate obtained in this systematic review is surely lower than the real one in the included studies, since, in some articles, patients were excluded due to membrane displacement [37] or very premature exposure [27], not taking into account these cases in the complication rate reported. Furthermore, in two of the studies, some exposed membranes are associated with graft failure, but it is not specified whether it is due to graft infection [24,39].

On the one hand, some of the included articles defend that the exposure of the titanium foil does not influence the success of the GBR [19,22,37,38], even if one of them sustains that the very early exposure favors the increase in width of the attached gingiva, unlike what happens with a later exposure [22]. On the other hand, two studies support that early exposure (before 14 days) has a worse prognosis than late exposure, with very poor bone gain [24,39]. 

From the results obtained, it appears that the survival rate of implants placed in regenerated bone is similar to that of implants placed in native bone [45].

This review is based on the scant scientific literature published on the matter so far, and, for the moment, it is the only existing systematic review, so the results obtained cannot be compared and cannot be given much value. Other limitations are the heterogeneity of the included studies, the small sample size of some of them and the lack of information regarding bone gain or membrane removal. For these reasons, a quantitative analysis could not be performed.

## 5. Conclusions

Based on the data presented above, titanium membranes in GBR should be considered as an incipient technique, versatile in terms of the type of bone defect to regenerate, in which there is still no evidence of the need of filling material and which is the most appropriate, that can better tolerate exposure than titanium meshes and titanium-reinforced non-resorbable membranes and that can be tailored to the patient’s bone defect.

More randomized clinical trials comparing occlusive titanium barriers and other types of membranes are necessary to obtain more robust data that allow us to reach solid conclusions regarding the predictability and complications rate associated with the use of titanium foils, and how to manage complications when they occur.

## Figures and Tables

**Figure 1 materials-13-05346-f001:**
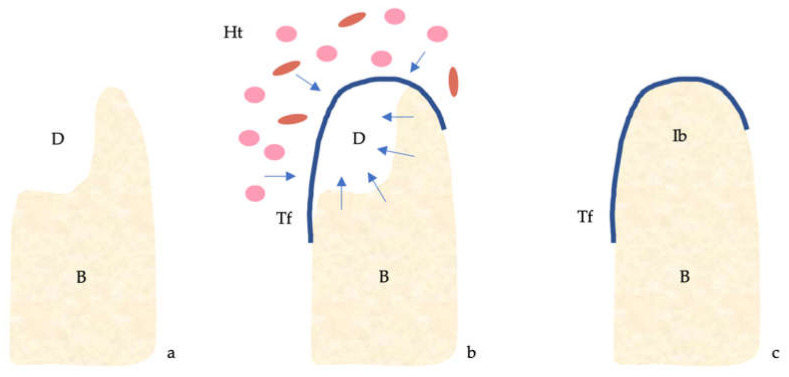
Guided bone regeneration (GBR) mechanism. (**a**) Bone defect. (**b**) The titanium barrier prevents the penetration of epithelial cells and fibroblasts and allows access to the defect of osteogenic and stem cells originating from the native bone. (**c**) Regeneration of the bone defect. Abbreviations: B, bone; D, defect; Ht, healing tissue; Ib, immature bone; Tf, titanium foil.

**Figure 2 materials-13-05346-f002:**
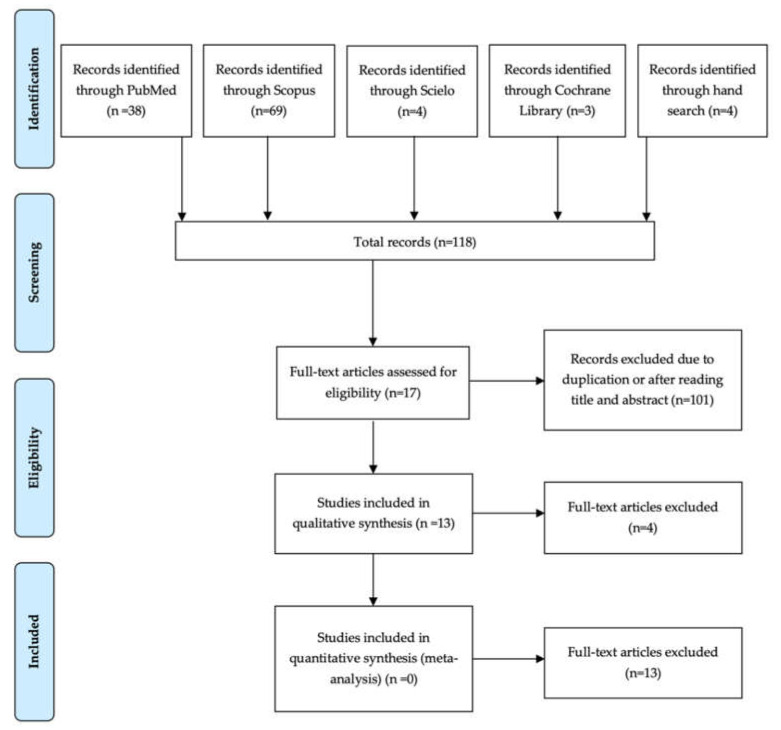
Preferred Reporting Items of Systematic Reviews and Meta-Analyses (PRISMA) flow diagram of selection process.

**Figure 3 materials-13-05346-f003:**
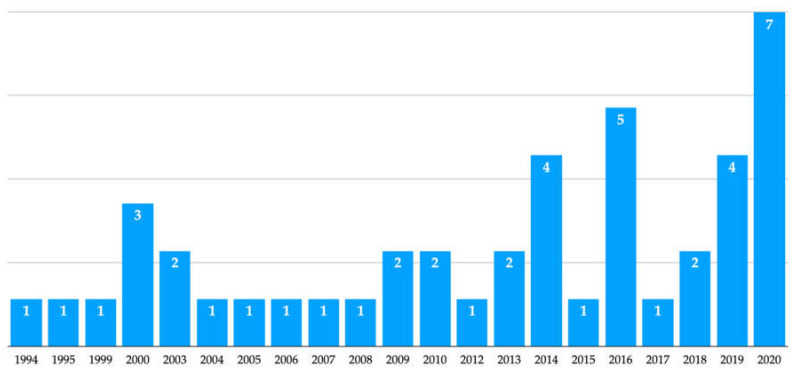
Results by year of the electronic search in PubMed.

**Table 1 materials-13-05346-t001:** Summary of the included studies.

Author	Type of Study	Level of Evidence	N (Titanium Foils)	Type of Defect	Filling Material	Membrane Removal (Months)	Bone Gain (mm)	Membrane Exposure (%)	Infection (%)
Kfir et al., 2007 [19]	Case series	Level 3	15	Post-extraction socket	PRF or PRF + TCP + HAP	NS	NS	47%	0%
Bassi et al., 2016 [22]	Case series	Level 3	13	HV	Allograft	6.35 ± 2.15	5.9 ± 2.1 V; 8.9 ± 3.5 H	38.5%	0%
Maeda et al., 2020 [25]	Case series	Level 3	15	H	Xenograft	21 days	8.02 ± 2.43 (P1); 8.71 ± 2.26 (P2); 9.00 ± 2.52 (P3)	100% (intentionally exposed)	0%
Perret et al., 2019 [26]	Case series	Level 3	6	HV	Xenograft	4	7.3 ± 2.2 (P1) 4.2 ± 1.2 (P2) V; 2.3 ± 1.0 H	100% (intentionally exposed)	0%
Engelke et al., 2014 [28]	Case report	Level 3	2	1HV; 1H	β-Tricalcium phosphate	3; NS	NS	0%	0%
Molly et al., 2006 [24]	Retrospective study	Level 2	11	HV	Blood clot	9–17	NS	45.5%	NS
Toygar et al., 2009 [27]	RCT	Level 2	16	Periodontal Intrabony Defect	Blood clot	1–1.5	NS	43.75%	NS
Gaggl et al., 1999 [31]	Case series	Level 3	42	Peri-implant Defect	Autograft + xenograft	NS	4.5 ± 0.2	21.43%	11.90%
Beltrán et al., 2013 [30]	Case report	Level 3	1	H	Xenograft	7	4 mm	0%	0%
Beltrán et al., 2014 [29]	Case series	Level 3	5	H	Allograft	6	2.3 (P1); 2.7 (P2); 2.9 (P3)	0%	0%
Khanna et al., 2016 [37]	RCT	Level 2	12	Periodontal Intrabony Defect	Blood clot	5–6 weeks	54.69% defect fill	33.33%	0%
Pinho et al., 2006 [38]	RCT	Level 2	10	Post-extraction socket	Blood clot (CS); autograft (TS)	Maximum 6 months	8.80 ± 2.93 (C); 8.40 ± 3.35 (T)	50%	0%
Watzinger et al., 2000 [39]	Case series	Level 3	112	Different type defects	Autograft and/or hydroxyapatite	4.6	NS	30%	NS

Abbreviations: C, control; CS, control socket; H, horizontal; HAP, hydroxyapatite; HV, horizontal and vertical; NS, not specified; P, point; PRF, platelet-rich fibrin; RCT, randomized clinical trial; T, test; TCP, tricalcium phosphate; TS, test socket; V, vertical.

**Table 2 materials-13-05346-t002:** Risk of bias across randomized clinical trials and the cohort study.

Bias Domain ROBINS-1	Molly et al. [24]	Bias Domain RoB 2	Khanna et al. [37]	Pinho et al. [38]	Toygar et al. [27]
Confounding	Low	Randomization process	Unclear	Unclear	Unclear
Selection of participants	High				
Classification of interventions	Low				
Deviations from intended interventions	Low	Deviations from intended interventions	Low	Low	Low
Missing data	Low	Missing data	Low	Low	Unclear
Measurement of outcomes	Low	Measurement of outcomes	Unclear	Low	Low
Selection of reported result	Unclear	Selection of reported result	Low	Low	Low
Overall bias	Unclear				

**Table 3 materials-13-05346-t003:** Implants survival and success rates.

Author	N Patients	N Implants	Survival Rate	Success Rate	Follow-Up
Kfir et al., 2007 [19]	8	9	-	-	-
Bassi et al., 2016 [22]	13	23	100%	82.6%	1 year
Perret et al., 2019 [26]	6	6	100%	100%	2 years
Engelke et al., 2014 [28]	2	3	100%	-	2 years
Molly et al., 2006 [24]	9	46	82.6%	-	6–9 years
Beltrán et al., 2013 [30]	1	1	100%	-	-
Total	39	88	96.5%	91.3%	1–9 years

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
