# Peer review of "Evaluation of Bone Gain and Complication Rates after Guided Bone Regeneration with Titanium Foils: A Systematic Review"

_materials, 2020, doi:10.3390/ma13235346_

Round 1
Reviewer 1 Report
This review article is a collection of data of 13 research articles that fell under the review's eligibility criteria. Unfortunately, this review does not provide full background to what extent titanium membranes are being studied. The conclusion that more clinical trials are required and that titanium membranes in GBR is incipient agreeable. However, this review article explains how the data were collected to write the review. It does not summarize the different methods and results of titanium foil usage. For example, if there were considerable bone gain or complications to the procedure, there should be an explanation of which procedures were used and what led to such conclusions, which this review article does not have.
Reviewer 2 Report
The paper reports a review on the bone regeneration techniques based on the titanium membranes. The MS can be published after the following minor revisions:
- In the Introduction, the authors should evidence that biocompatible materials for bone regeneration purposes were recently fabricated by combining biopolymers and natural nanoparticles, such as nanoclays. This sentence should be supported by corresponding recent reviews [Carbohydrate Polymers 245 (2020) 116502; Therapeutic Delivery 9, 4, 2018, 287-301]
- The distribution per year of the published articles within the last decades should be presented. The electronic search criteria should be the same of those used in the submitted MS.
- Paragraph 3.3. The authors stated "According to the Strength of Recommendation Taxonomy (SORT) criteria, four of the studies included obtained a Level 2 of evidence while the others obtained a Level 3 ". I suggest to report and discuss the SORT criteria in the text of the paragraph 3.3.
Reviewer 3 Report
The work entitled “Titanium foils: bone gain and complications. Systematic review” reported by Millan et al has been reviewed. This work systematically reviewed the work published relation to the titanium foils in bone gain and its complication. However, it needs further addition to improve the quality of the review analysis.
Specific comments:
Throughout the MS need the exclusive English correction. Particularly the tile of the article needs to rephrased.
The basic and fundamental information need to be included in the introduction section. For example. What is the mechanism beyond the GBR? Role and utilization of Titanium foils ?
In the line no 48-49: check the spelling mistake and rephase the sentence “ However, due to the presence of erythrocytes in the blood, it has a greater capacity to generate thrombin and activate platelets tan these concentrates”
IN line 92 explain ER,EJ?
This review seems to be a short communication not the compelte systematic review need to add the more additional detail information for the sections such as 3.4, 3.5, 3.6
This article not consisted with conclusion and take-home message.
Need to added the more pectoral representation for strengthen the article quality.
Reviewer 4 Report
In this manuscript (materials-999910), authors have provided a systematic review on the efficacy of titanium membranes for bone gain and respective complication rate. Specifically, this study focuses on analyzing the success rate of guided-bone regeneration techniques as using titanium membranes by including most recent published case reports, cohort studies, and clinical trials in humans. This focused review is interesting and timely written for reaching more valid conclusions for future research directions. While gone throughout the manuscript, I have found that this review article has good potential to be published in this journal.
Here, I have reviewed this manuscript thoroughly and found all sections, especially materials & methods and results & discussion, are nicely designed and evaluated. However, more specifically, this manuscript can be improved for better understanding, more specifically as follows:
1) In lines 58-64, authors should provide 2 or 3 figures or schematic illustrations of these titanium barriers to present their efficacy for guided bone tissue regeneration.
2) Conclusion section should be described separately, with advantages, limitations, and future research directions in this area.
In my opinion, this manuscript can be accepted in its current form for publication.
Reviewer 5 Report
Dear authors,
thanks for having provided such an interesting manuscript.
Here you can find my comments:
- Line 41: Please give appropriate citation to the sentence about the "gold standard";
- Line 70: "according" is repeated twice, please correct;
- Line 72: please report when the protocol is located and if it can be accessed;
- Line 76: "complications" has to be correct in "complication";
- Line 94: About manual search a list of the consulted journals (with specific issues or time intervals) has to be provided;
- Line 96: It has to be reported if the words used for the database search have been used as MeSH terms of free text words;
- Line 104: "The data was" has to be corrected in "the data were";
- Line 105: It is reported that in case of disagreement between the two reviewers a third reviewer was consulted but in parentheses 2 names are provided, could you please explain better?;
- Line 105: About data collection, how was this done? Using specially designed table?;
- Line 113-118: You reported that you included also case report and case series but how did you evaluate them for the eventual risk of bias?; They evaluation is missing, please provide;
- Line 121-127: Please give reasons for not having performed a quantitative analysis of data;
- The first two rows of figure 1 are cropped, please fix them;
- Line 158: "de" must be changed into "the";
- Line 170-175: Final global judgments for risk of bias are not given, please provide;
- Table 1 column 2 row 6: "studi" must be changed in "study";
- Line 278-180: I do not agree with the affirmation especially since you previously reported in line 242-243 that "Based on the included articles, there is no evidence to believe that a filling material is better than another or even blood clot, in combination with occlusive titanium barriers". Please check what is reported on line 278-280;
Regards
Round 2
Reviewer 3 Report
Accept
Reviewer 5 Report
Dear authors,
the changes you made are all satisfactory.
The article can be accepted in its present form.
Regards